# A Johnson–Lindenstrauss Framework for Randomly Initialized CNNs

**Ido Nachum, Jan Hązła, Michael Gastpar**
School of Computer and Communication Sciences
École Polytechnique Fédérale de Lausanne
1015 Lausanne, Switzerland
⟨forename.surname⟩@epfl.ch

**Anatoly Khina**
School of Electrical Engineering
Tel Aviv University
Tel Aviv 6997801, Israel
anatolyk@eng.tau.ac.il

## Abstract

How does the geometric representation of a dataset change after the application of each randomly initialized layer of a neural network? The celebrated Johnson–Lindenstrauss lemma answers this question for linear fully-connected neural networks (FNNs), stating that the geometry is essentially preserved. For FNNs with the ReLU activation, the angle between two inputs contracts according to a known mapping. The question for non-linear convolutional neural networks (CNNs) becomes much more intricate. To answer this question, we introduce a geometric framework. For linear CNNs, we show that the Johnson–Lindenstrauss lemma continues to hold, namely, that the angle between two inputs is preserved. For CNNs with ReLU activation, on the other hand, the behavior is richer: The angle between the outputs contracts, where the level of contraction depends on the nature of the inputs. In particular, after one layer, the geometry of natural images is essentially preserved, whereas for Gaussian correlated inputs, CNNs exhibit the same contracting behavior as FNNs with ReLU activation.

## 1 Introduction

Neural networks have become a standard tool in multiple scientific fields, due to their success in classification (and estimation) tasks. Conceptually, this success is achieved since better representation is allowed by each subsequent layer until linear separability is achieved at the last (linear) layer. Indeed, in many disciplines involving real-world tasks, such as computer vision and natural language processing, the training process is biased toward these favorable representations. This bias is a product of several factors, with the neural-network initialization playing a pivotal role (Sutskever et al., 2013). Therefore, we concentrate in this work on studying the initialization of neural networks, with the following question guiding this work.

*How does the geometric representation of a dataset change after the application of each randomly initialized layer of a neural network?*

To answer this, we study how the following two geometrical quantities change after each layer.

$$\langle x, y \rangle \qquad \text{The scalar \textit{inner product} between vectors } x \text{ and } y.[1]$$
$$\rho := \frac{\langle x, y \rangle}{\|x\|\|y\|} \qquad \text{The \textit{cosine similarity} (or simply \textit{similarity}) between vectors } x \text{ and } y.$$

The similarity $\rho \in [-1, 1]$ between $x$ and $y$ equals $\rho = \cos(\theta)$, where $\theta$ is the angle between them.

Consider first one layer of a fully connected neural network (FNN) with an identity activation (linear FNN) that is initialized by independent identically distributed (i.i.d.) Gaussian weights with mean zero and variance $1/N$, where $N$ is the number of neurons in that layer. This random linear FNN induces an isometric embedding of the dataset, namely, the similarity $\rho_{\text{in}}$ between any two inputs,

---

[1]We mean here a vector in a wider sense: $x$ and $y$ may be matrices and tensors (of the same dimensions). In this situation, the standard inner product is equal to the vectorization thereof: $\langle x, y \rangle = \langle \text{vec}(x), \text{vec}(y) \rangle$.

$x_{\text{in}}$ and $y_{\text{in}}$, is preserved together with their norm:

$$\rho_{\text{out}} \approx \bar{\rho}_{\text{out}} = \rho_{\text{in}},$$

where $\rho_{\text{out}}$ is the similarity between the resulting random (due to the multiplication by the random weights) outputs, $X_{\text{out}}$ and $Y_{\text{out}}$ (respectively), and $\bar{\rho}_{\text{out}}$ is the mean output similarity defined by

$$\rho_{\text{out}} := \frac{\langle X_{\text{out}}, Y_{\text{out}} \rangle}{\|X_{\text{out}}\| \, \|Y_{\text{out}}\|}, \qquad \text{and} \qquad \bar{\rho}_{\text{out}} := \frac{\mathbb{E}\left[\langle X_{\text{out}}, Y_{\text{out}} \rangle\right]}{\sqrt{\mathbb{E}\left[\|X_{\text{out}}\|^2\right] \mathbb{E}\left[\|Y_{\text{out}}\|^2\right]}}. \tag{1}$$

The proof of this isometric relation between the input and output similarities follows from the celebrated Johnson–Lindenstrauss lemma (Johnson and Lindenstrauss, 1984; Dasgupta and Gupta, 1999). This lemma states that a random linear map of dimension $N$ preserves the distance between any two points up to an $\epsilon > 0$ contraction/expansion with probability at least $1 - \delta$ for all $N > c \log(1/\delta)/\epsilon^2$ for an absolute constant $c$. In the context of randomly initialized linear FNNs, this result means that, for a number of neurons $N$ that satisfies $N > c \log(1/\delta)/\epsilon^2$,

$$\mathbb{P}\left(|\rho_{\text{out}} - \rho_{\text{in}}| < \epsilon\right) \geq 1 - \delta$$

So, conceptually, the Johnson–Lindenstrauss lemma studies how inner products (or geometry) change, in expectation, after applying a random transformation and how well an average of these random transformations is concentrated around the expectation. This is the exact setting of randomly initialized neural networks. The random transformations consist of a random projection (multiplying the dataset by a random matrix) which is followed by a non-linearity.

Naturally, adding a non-linearity complicates the picture. Let us focus on the case where the activation function is a rectified linear unit (ReLU). That is, consider a random fully-connected layer with ReLU activation initialized with i.i.d. zero-mean Gaussian weights and two different inputs. For this case, Cho and Saul (2009), Giryes et al. (2016), and Daniely et al. (2016) proved that[2]

$$\rho_{\text{out}} \approx \bar{\rho}_{\text{out}} = \frac{\sqrt{1 - \rho_{\text{in}}^2} + \left(\pi - \cos^{-1}(\rho_{\text{in}})\right) \rho_{\text{in}}}{\pi}. \tag{2}$$

Following Daniely et al. (2016), we refer to the resulting function in (2) of $\bar{\rho}_{\text{out}}$ in $\rho_{\text{in}}$ as the *dual activation* of the ReLU; this function is represented in figure 1 by the yellow curve.

One may easily verify that the dual activation (2) of the ReLU satisfies $\rho_{\text{out}} > \rho_{\text{in}}$ for $\rho_{\text{in}} \neq 1$, meaning that it is a *contraction*. Consequently, for deep FNNs, which comprise multiple layers, random initialization results in a *collapse of all inputs* with the same norm (a sphere) *to a single point* at the output of the FNN (equivalently, the entire dataset collapses to a single straight line).

Intuitively, this collapse is an unfavorable starting point for optimization. To see why, consider the gradient $\nabla_{w_j^{(i)}}$ of the weight $w_j^{(i)}$ of some neuron $j$ in a deep layer $i$ in a randomly initialized ReLU FNN. By the chain rule (backpropagation), this gradient is proportional to the output of the previous layer $a^{(i-1)}$ for the corresponding input, i.e., it holds that $\nabla_{w_j^{(i)}} \propto a^{(i-1)}$. If the collapse is present already at layer $i$, this output is essentially proportional to a fixed vector $a^*$. But this implies that, in the gradient update, the weights of the deep layer will move roughly along a straight line which would impede, in turn, the process of achieving linear separability.

Indeed, it is considered that for a FNN to train well, its input–output Jacobian needs to exhibit *dynamical isometry* upon initialization (Saxe et al., 2014). Namely, the singular values of the Jacobian $\partial x_{\text{out}}/\partial x_{\text{in}}$ must be concentrated around 1, where $x_{\text{out}}$ and $x_{\text{in}}$ denote the input and output of the FNN, respectively. If the dataset collapses to a line, $x_{\text{out}}$ is essentially invariant to $x_{\text{in}}$ (up to a change in its norm), suggesting that the singular values of $\partial x_{\text{out}}/\partial x_{\text{in}}$ are close to zero. Therefore, randomly initialized FNNs exhibit the *opposite* behavior from dynamical isometry and hence do not train well.

---

[2]The results in (Cho and Saul, 2009) and (Daniely et al., 2016) were derived assuming unit-norm vectors $\|x_{\text{in}}\| = \|y_{\text{in}}\| = 1$. The result here follows by the homogeneity of the ReLU activation function: $\text{R}(\alpha x) = \alpha \, \text{R}(x)$ for $\alpha \geq 0$, and ergodicity, assuming multiple filters are applied.

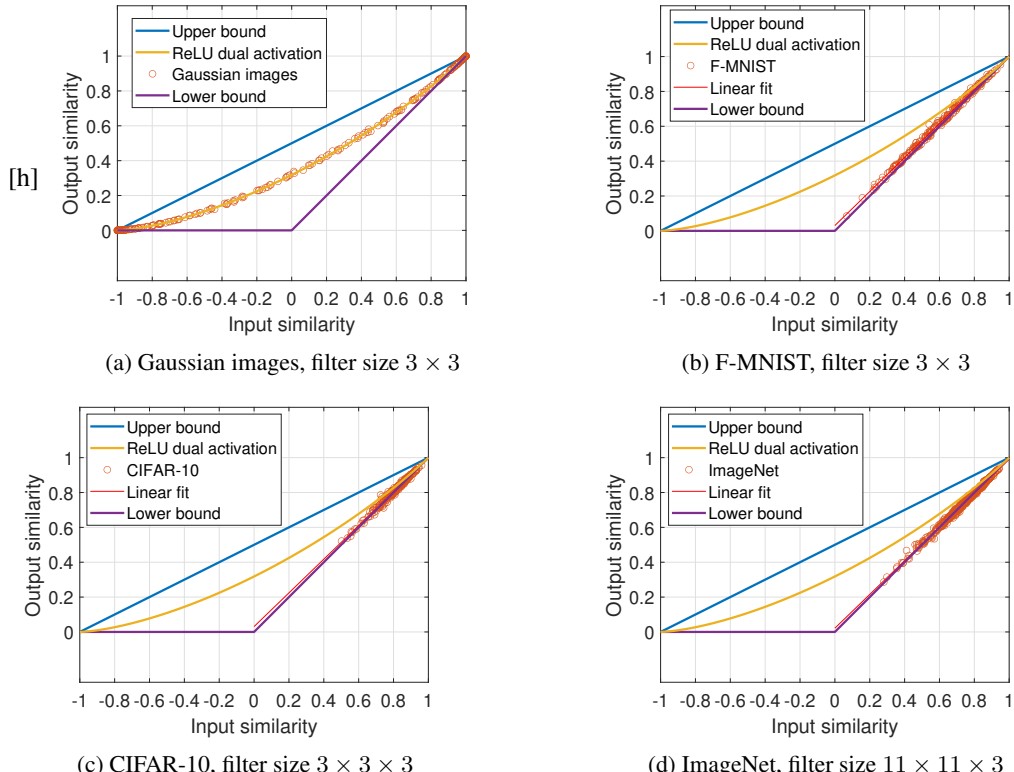

(a) Gaussian images, filter size $3 \times 3$        (b) F-MNIST, filter size $3 \times 3$

(c) CIFAR-10, filter size $3 \times 3 \times 3$        (d) ImageNet, filter size $11 \times 11 \times 3$

Figure 1: Input and output cosine similarities of a single randomly initialized convolutional layer with 100 filters. Each red circle in the figures represents a random pair of images chosen from the corresponding dataset. 200 pairs were sampled in each figure.

## 1.1 OUR CONTRIBUTION

Our main interest lies in the following question.

*Does the contraction observed in randomly initialized ReLU FNNs carry over to convolutional neural networks (CNNs)?*

As we will show, qualitatively, the answer is yes. However, quantitatively, the answer is more subtle as is illustrated in figure 1. In this figure, the similarity between pairs of inputs sampled at random from standard natural image datasets—Fashion MNIST (F-MNIST), CIFAR-10, and ImageNet—are displayed against the corresponding output of a randomly initialized CNN layer. For these datasets, clearly $\rho_{\text{out}} \approx \rho_{\text{in}}$ meaning that the relation of ReLU FNNs (2)—represented in figure 1 by the yellow curve—breaks down.

That said, for inputs consisting of i.i.d. zero-mean Gaussians (and filters comprising i.i.d. zero-mean Gaussian weights as before) with a Pearson correlation coefficient $\rho$ between corresponding entries (and independent otherwise), the relation in (2) between $\bar{\rho}_{\text{out}}$ and $\rho_{\text{in}}$ of ReLU FNNs does hold for ReLU CNNs as well, as illustrated in figure 1a.

This dataset-dependent behavior, observed in figure 1, suggests that, in contrast to randomly-initialized FNNs which behave according to (2), randomly-initialized CNNs exhibit a richer behavior: $\bar{\rho}_{\text{out}}$ does not depend only on $\rho_{\text{in}}$ but on the inputs $x_{\text{in}}$ and $y_{\text{in}}$ themselves. Therefore, in this work, we characterize the behavior of $\bar{\rho}_{\text{out}}$ after applying one layer in randomly initialized CNNs.

We start by considering randomly initialized CNNs with general activation functions. We show in theorem 1 that the expected (over the filters) inner product $\mathbb{E}\left[\langle X_{\text{out}}, Y_{\text{out}}\rangle\right]$ and the mean similarity $\bar{\rho}_{\text{out}}$ depend on $x_{\text{in}}$ and $y_{\text{in}}$ (and not just $\langle x_{\text{in}}, y_{\text{in}}\rangle$) by extending the dual-activation notion of Daniely et al. (2016). In theorem 2, we further prove that, by taking multiple independent filters, $\langle X_{\text{out}}, Y_{\text{out}}\rangle$ and $\rho_{\text{out}}$ of (1) concentrate around $\mathbb{E}\left[\langle X_{\text{out}}, Y_{\text{out}}\rangle\right]$ and $\bar{\rho}_{\text{out}}$, respectively.

We then specialize these results to linear CNNs (with identity activation) and derive a convolution-based variant of the Johnson–Lindenstrauss lemma that shows that $\langle X_{\text{out}}, Y_{\text{out}} \rangle \approx \langle x_{\text{in}}, y_{\text{in}} \rangle$ and $\rho_{\text{out}} \approx \rho_{\text{in}}$ for linear CNNs, both in expectation ($\bar{\rho}_{\text{out}} = \rho_{\text{in}}$ for the latter) and with high probability.

For randomly initialized ReLU CNNs, we derive the following tight upper and lower bounds for $\bar{\rho}_{\text{out}}$ in terms of $\rho_{\text{in}}$ in theorem 3:

$$\max\{\rho_{\text{in}}, 0\} \le \bar{\rho}_{\text{out}} \le \frac{1 + \rho_{\text{in}}}{2}. \tag{3}$$

These bounds imply, in turn, that for $\rho_{\text{in}} \ne 1$ each ReLU CNN layer is contracting. In theorem 4 we prove that $\bar{\rho}_{\text{out}}$ for random Gaussian data satisfies the relation (2), in accordance with figure 1a.

To explain the (almost) isometric behavior of CNNs for natural images (figure 1), we note that many natural images consist of large, relative to the filter size, approximately monochromatic patches. This observation leads to a simple model of black and white (binary) images with "large patches". To describe this model mathematically, we define a notion of a shared boundary between two images in definition 2, and model large patches by bodies whose area is large compared to the shared boundary. We prove that $\bar{\rho}_{\text{out}} \approx \rho_{\text{in}}$ for this model, meaning that the lower bound in (3) is in fact tight.

## 1.2 RELATED WORK

In this paper, we study how various inputs are embedded by randomly initialized convolutional neural networks. Neural tangent kernel (NTK) (Jacot et al., 2018) is a related line of work. This setting studies the infinite width limit (among other assumptions) in which one can consider neural network training as regression over a fixed kernel; this kernel is the NTK. There are two factors that affect the calculation of the NTK: The embedding of the input space at initialization and the gradients at initialization. In this paper we study the first one.

Arora et al. (2019), and Bietti and Mairal (2019) give expressions for the NTK and the convolutional NTK (CNTK). Theorem 1 may be implicitly deduced from those expressions. Arora et al. (2019) provide concentration bounds for the NTK of fully connected networks with finite width. Bietti and Mairal (2019) derive smoothness properties for NTKs, e.g., upper bounds on the deformation induced by the NTK in terms of the initial Euclidean distance between the inputs. A related approach to NTK is taken in (Bietti, 2021) where convolutional kernel networks (CKN) are used.

Standard initialization techniques use the Glorot initializtion (Glorot and Bengio, 2010) and He initialization (He et al., 2015). Both were introduced to prevent the gradients from exploding/vanishing. On a similar note, Hanin and Rolnick (2018) discuss how to prevent exploding/vanishing mean activation length—which corresponds to the gradients to some degree—in FNN with and without skip connections. For a comprehensive review on other techniques see (Narkhede et al., 2021-06-28).

Schoenholz et al. (2017); Poole et al. (2016); Yang and Schoenholz (2017) study the initialization of FNNs using mean-field theory, in a setting where the width of the network is infinite. They demonstrate that for some activation functions there exists an initialization variance such that the network does not suffer from vanishing/exploding gradients. In this case, the network is said to be initialized at the edge of chaos.

As mentioned earlier, Saxe et al. (2014) introduced a stronger requirement than that of moderate gradients at initialization, in the form of *dynamical isometry*. for linear FNNs, they showed that orthogonal initialization achieves dynamical isometry whereas Gaussian i.i.d. initialization does not.

For non-linear FNNs, Pennington et al. (2017) show that the hyperbolic tangent (tanh) activation can achieve dynamical isometry while ReLU FNNs with Gaussian i.i.d. initialization or orthogonal initialization cannot. In contrast, Burkholz and Dubatovka (2019) show that ReLU FNNs achieve dynamical isometry by moving away from i.i.d. initialization. And lastly, Tarnowski et al. (2019) show that residual networks achieve dynamical isometry over a broad spectrum of activations (including ReLU) and initializations.

Xiao et al. (2018) trained tanh (and not ReLU) CNNs with 10000 layers that achieve dynamical isometry using the delta-orthogonal initialization. Similarly, Zhang et al. (2019) trained residual CNNs with 10000 layers without batch normalization using fixup-initialization that prevents exploding/vanishing gradients.

## 2 SETTING AND DEFINITIONS

The output $z \in \mathbb{R}^{n \times n}$ of a single filter of a CNN is given by

$$z = \sigma(F * x)$$

where $x \in \mathbb{R}^{n \times n \times d}$ is the input; $F \in \mathbb{R}^{r \times r \times d}$ is the filter (or kernel) of this layer; and $\sigma : \mathbb{R} \to \mathbb{R}$ is the activation function of this layer and is understood to apply entrywise when applied to tensors. The dimensions satisfy $n, d, r \in \mathbb{N}, r \leq n$. The first two dimensions of $x$ and $z$ are the image dimensions (width and height), while the third dimension is the number of channels, and is most commonly equal to either 1 for grayscale images and 3 for RGB color images; $r$ is the dimension of the filter; '$*$' denotes the cyclic convolution operation:

$$(F * x)_{uv} = \sum_{i \in \mathbb{Z}_r, j \in \mathbb{Z}_r, k \in \mathbb{Z}_d} F_{ijk} \cdot x_{u-i,v-j,k} \tag{4}$$

where the subtraction operation in the arguments is the subtraction operation modulo $n$. In this work, we will concentrate on the ReLU activation function which is given by $\sigma(x) = \max\{0, x\}$. Typically, a single layer of a CNN consists of multiple filters $F_1, \ldots, F_N$.

The following definition is an adaptation of the dual activation of Daniely et al. (2016) to CNNs.

**Definition 1** (Dual activation). The dual activation function $\widehat{\sigma} : \mathbb{R}^n \times \mathbb{R}^n \times \mathbb{R} \to \mathbb{R}$ of an activation function $\sigma$ is defined as

$$\widehat{\sigma}(x, y, \nu) = \mathbb{E}\left[\sigma(\nu X) \cdot \sigma(\nu Y)\right]$$

for input vectors $x, y \in \mathbb{R}^n$, and a parameter $\nu > 0$, where

$$\begin{pmatrix} X \\ Y \end{pmatrix} \sim \mathcal{N}\left(\begin{pmatrix} 0 \\ 0 \end{pmatrix}, \begin{pmatrix} \|x\|^2 & \langle x, y \rangle \\ \langle x, y \rangle & \|y\|^2 \end{pmatrix}\right). \tag{5}$$

For tensors $x, y \in \mathbb{R}^{n \times n \times d}$, the dual activation $\widehat{\sigma}$ is defined as in (5) with the inner products and norms replaced by the standard inner product between tensors:

$$\langle x, y \rangle = \sum_{i \in \mathbb{Z}_n, j \in \mathbb{Z}_n, k \in \mathbb{Z}_d} x_{ijk} \cdot y_{ijk}, \qquad \|x\| = \sqrt{\langle x, x \rangle}.$$

In the special case of an activation that is homogeneous, i.e., satisfying $\sigma(cx) = c\sigma(x)$ for $c \geq 0$, we use an additional notation for $-1 \leq \rho \leq 1$, with some abuse of notation:

$$\nu_\sigma^2 := \mathbb{E}\left[\sigma(X)^2\right], \quad \widehat{\sigma}(\rho) := \frac{\mathbb{E}[\sigma(X) \cdot \sigma(Y)]}{\nu_\sigma^2}, \quad \begin{pmatrix} X \\ Y \end{pmatrix} \sim N\left(\begin{pmatrix} 0 \\ 0 \end{pmatrix}, \begin{pmatrix} 1 & \rho \\ \rho & 1 \end{pmatrix}\right). \tag{6}$$

This is the dual activation function of Daniely et al. (2016). Note that, by this definition, $\widehat{\sigma}(1) = 1$.

We will be mostly interested in the ReLU activation, denoted by R, for which the following holds (see, e.g., (Daniely et al., 2016, Table 1 and Section C of supplement))

$$\nu_R^2 := 1/2, \quad \widehat{R}(\rho) := \frac{\mathbb{E}[R(X) \cdot R(Y)]}{\nu_R^2} = \frac{\sqrt{1 - \rho^2} + (\pi - \cos^{-1}(\rho))\rho}{\pi}.$$

## 3 MAIN RESULTS

In this section, we present the main results of this work. **The proofs of all the results in this section may be found in the appendix.**

**Theorem 1.** *Let $x, y \in \mathbb{R}^{n \times n \times d}$ be inputs to a convolution filter $F \in \mathbb{R}^{r \times r \times d}$ with $r \leq n$ and activation function $\sigma$ such that the entries of $F$ are i.i.d. Gaussian with variance $\nu^2$. Then,*

$$\mathbb{E}\left[\langle\sigma(F * x), \sigma(F * y)\rangle\right] = \sum_{i,j \in \mathbb{Z}_n} \widehat{\sigma}\left([x]_{ij}^r, [y]_{ij}^r, \nu\right). \tag{7}$$

*In particular, for a homogeneous activation function we have*

$$\mathbb{E}\left[\langle\sigma(F * x), \sigma(F * y)\rangle\right] = \nu^2 \cdot \nu_\sigma^2 \sum_{i,j \in \mathbb{Z}_n} \left\|[x]_{ij}^r\right\| \left\|[y]_{ij}^r\right\| \widehat{\sigma}\left(\rho_{ij}\right), \tag{8}$$

$$\mathbb{E}\left[\sigma(F * x)^2\right] = \nu^2 \cdot \nu_\sigma^2 \cdot r^2 \|x\|^2 , \tag{9}$$

*where $\rho_{ij} = \frac{\langle[x]_{ij}^r, [y]_{ij}^r\rangle}{\|[x]_{ij}^r\| \cdot \|[y]_{ij}^r\|}$ and $\nu_\sigma^2$ has been defined in* (6).

In particular, due to (9), for homogeneous activations we can choose $\nu = 1/(\nu_\sigma r)$ and get $\mathbb{E}\left[\sigma(F * x)^2\right] = \|x\|^2$. That is, the norms are preserved in expectation for every input $x$.

Note that theorem 1 may be deduced as a special case from existing more general formulas; see, e.g., Arora et al. (2019); Bietti and Mairal (2019). Nevertheless, it is an important starting point for us.

While theorem 1 holds in the mean sense, it does not hold for a specific realization of a single filter $F$, in general. The next theorem, whose proofs relies on the notions of sub-Gaussian and sub-exponential RVs and their concentration of measure, states that theorem 1 holds approximately for a large enough number of applied filters.

We apply basic, general concentration bounds on the activation. There exist sharper bounds in special cases, e.g., in the context of cosine similarity and ReLU, see (Buchanan et al., 2021, app. E).

**Theorem 2.** *Let $x, y \in \mathbb{R}^{n \times n \times d}$ be inputs to $N$ filters $F_1, \ldots, F_N \in \mathbb{R}^{r \times r \times d}$ with $r \leq n$ such that*

$$\max_{i,j \in \mathbb{Z}_n} \left\|[x]_{ij}^r\right\| \leq R, \qquad\qquad \max_{i,j \in \mathbb{Z}_n} \left\|[y]_{ij}^r\right\| \leq R, \tag{10}$$

*all the entries of all the filters are i.i.d. Gaussian with zero mean and variance $\nu^2$, and a Lipschitz continuous activation function $\sigma$ with a Lipschitz constant $L$ and satisfying $\sigma(0) = 0$. Then, for $\delta > 0$, however small,*

$$\mathbb{P}\left(\left|\frac{1}{N}\sum_{\ell=1}^{N}\langle\sigma(F_\ell * x), \sigma(F_\ell * y)\rangle - \sum_{i,j=1}^{n}\widehat{\sigma}\left([x]_{ij}^r, [y]_{ij}^r, \nu\right)\right| \geq \epsilon\right) \leq \delta \tag{11}$$

*for $N > \max\left(K, K^2\right)\log\frac{2n^2}{\delta}$, where $K = D\nu^2 L^2 R^2 n^2/\epsilon$ and $D > 0$ is an absolute constant.*

**Remark 1.** Clearly, $\|[x]_{ij}^r\|$ and $\|[y]_{ij}^r\|$ in (10) may be bounded by $\|x\|$ and $\|y\|$, respectively. However, since $r$ is typically much smaller than $n$, we prefer to state the result in terms of the norms of $[x]_{ij}^r$ and $[y]_{ij}^r$.

Theorem 2 states a concentration result in terms of the inner product. We now give a parallel result for homogeneous activations in terms of the cosine similarity $\rho_{\text{out}}$. To that end, consider a homogeneous activation $\sigma$ and $N$ random filters $F_1, \ldots, F_N$; for these filters, the quantities of (1) for two inputs $x$ and $y$ are equal to

$$\rho_{\text{out}} = \frac{\sum_{\ell=1}^{N}\langle\sigma\left(F_\ell * x\right), \sigma\left(F_\ell * y\right)\rangle}{\sqrt{\sum_{\ell=1}^{N}\|\sigma\left(F_\ell * x\right)\|^2}\sqrt{\sum_{\ell=1}^{N}\|\sigma\left(F_\ell * y\right)\|^2}}, \quad \bar{\rho}_{\text{out}} = \frac{\sum_{i,j \in \mathbb{Z}_n} \|[x]_{ij}^r\| \cdot \|[y]_{ij}^r\|\widehat{\sigma}\left(\rho_{ij}\right)}{r^2\|x\| \cdot \|y\|}, \tag{12}$$

where $\rho_{ij}$ is defined in theorem 1. Applying theorem 2 to these quantities, yields the following.

**Corollary 1.** *Let $x, y$ be inputs to $N$ filters $F_1, \ldots, F_N$ such that*

$$\max_{i,j \in \mathbb{Z}_n} \frac{\|[x]_{ij}^r\|}{\|x\|} \leq R , \qquad \max_{i,j \in \mathbb{Z}_n} \frac{\|[y]_{ij}^r\|}{\|y\|} \leq R ,$$

*all the entries of the filters are i.i.d. Gaussian with zero mean and variance $\nu^2$, and a Lipschitz homogeneous activation $\sigma$ with Lipschitz constant $L$. Then, for $0 < \epsilon \leq 1/10$ and $\delta > 0$,*

$$\mathbb{P}\left(|\rho_{\text{out}}(x,y) - \bar{\rho}_{\text{out}}(x,y)| \geq \epsilon\right) \leq \delta$$

*for $N > \max(K, K^2)\log\frac{6n^2}{\delta}$, where $K = \frac{DL^2R^2n^2}{\epsilon\nu_\sigma^2 r^2}$ and $D > 0$ is an absolute constant.*

**Remark 2.** To make sense out of the constants in corollary 1, we point out that applying it to inputs $x, y \in \{\pm 1\}^{n \times n \times d}$ (so that $\|x\|^2 = \|y\|^2 = dn^2$ and $\|[x]_{ij}^r\|^2 = \|[y]_{ij}^r\|^2 = dr^2$) results in $K = \frac{DL^2}{\epsilon \nu_\sigma^2}$, which does not depend on $n$ or $\nu^2$. There remains a $\log n^2$ factor that increases with $n$.

In the remainder of this section, we derive an analogous result to the Johnson–Lindenstrauss lemma (Johnson and Lindenstrauss, 1984; Dasgupta and Gupta, 1999) in section 3.1 for random *linear* CNNs, namely, CNNs with identity activation functions. We then move to treating random CNNs with ReLU activation functions: We derive upper and lower bounds on the inner product between the outputs in section 3.2, and prove that they are achievable for Gaussian inputs in section 3.3, and for large convex-body inputs in section 3.4, respectively.

## 3.1 LINEAR CNN

Theorem 1 yields a variant of the Johnson–Lindenstrauss lemma where the random projection is replaced with random filtering, that is, by applying a (properly normalized) random filter, the inner product is preserved (equivalently, the angle).

**Lemma 1.** *Consider the setting of theorem 1 with $\nu = 1/r$. Then, $\mathbb{E}\left[\langle F * x, F * y \rangle\right] = \langle x, y \rangle$.*

**Lemma 2.** *Consider the setting of theorem 2 with $\nu = 1/r$ (and $\sigma(x) \equiv x$). Then,*

$$\mathbb{P}\left(\left| \frac{1}{N} \sum_{\ell=1}^{N} \langle F_\ell * x, F_\ell * y \rangle - \langle x, y \rangle \right| \geq \epsilon \right) \leq \delta$$

*for $N > \max\left(K, K^2\right) \log \frac{2n^2}{\delta}$, where $K = \frac{DR^2 n^2}{r^2 \epsilon}$ and $D > 0$ is an absolute constant.*

Related result appears in Krahmer et al. (2014) (theorem 1.1) which shows norm preservation of sparse vectors for convolutional operators with a filter dimension that equals the input dimension.

## 3.2 CNN WITH ReLU ACTIVATION

The following theorem implies that every layer of a neural network is contracting in expectation (over $F$). That is, the expected inner product between any two data points will get larger with each random CNN layer. We also develop an upper bound on the new correlation.

**Theorem 3.** *Consider the setting of theorem 1 with a ReLU activation function and variance $\nu = 2/r^2$, for inputs $x$ and $y$ with similarity $\rho$. Then,*

$$\max\{\langle x, y \rangle, 0\} \leq \mathbb{E}\left[\langle \mathrm{R}(F * x), \mathrm{R}(F * y) \rangle\right] \leq \|x\| \, \|y\| \frac{1+\rho}{2}, \tag{13}$$

**Remark 3.** All the bounds in theorem 3 are tight. For simplicity, we present examples where the unit vectors $x, y$ are flat, i.e., they are not tensors and $r = 1$. Similar examples can be drawn for tensor inputs and $r > 1$. The upper bound is realized for $x, y \in \mathbb{R}^2$ of the form $x = \left(\sqrt{\frac{1+\rho}{2}}, \sqrt{\frac{1-\rho}{2}}\right)$, $y = \left(\sqrt{\frac{1+\rho}{2}}, -\sqrt{\frac{1-\rho}{2}}\right)$. Similarly, $x, y \in \mathbb{R}^3$ with $x = (\sqrt{1-\rho}, \sqrt{\rho}, 0)$, $y = (0, -\sqrt{\rho}, \sqrt{1-\rho})$ satisfy $\langle x, y \rangle = -\rho$ and $\mathbb{E}\left[\sigma(F * x)\sigma(F * y)\right] = 0$ for $r = 1$ and $0 \leq \rho \leq 1$. Finally, we can take $x = (\sqrt{1-\rho}, \sqrt{\rho}, 0)$ and $y = (0, \sqrt{\rho}, \sqrt{1-\rho})$ to obtain $\langle x, y \rangle = \mathbb{E}\left[\sigma(F * x)\sigma(F * y)\right] = \rho$.

These examples are illustrated in figure 3.

**Remark 4.** Although the lower bound is tight, in most typical scenarios it will be strict, so, in expectation, the convolutional layers will *contract* the dataset after application of enough layers. To see why, observe (23c); equality is achieved iff $\rho_{ij} = 1 \vee [x]_{ij}^r = 0 \vee [y]_{ij}^r = 0$. For real data sets this will never hold over all $(i, j)$. This can be observed in figures 1b, 1c, and 1d where the linear fit has a slope of 0.97.

## 3.3 GAUSSIAN INPUTS

The following result shows that the behavior reported by (Daniely et al., 2016) [and illustrated in figure 1a by the orange curve] for FNNs holds for CNNs with Gaussian correlated inputs; this is illustrated in figure 1a. For simplicity, we state the result for $d = 1$ ($n \times n$ inputs instead of $n \times n \times d$); the extension to $d > 1$ is straightforward.

**Theorem 4.** *Let $X, Y \in \mathbb{R}^{n \times n}$ be zero-mean jointly Gaussian with the following auto- and cross-correlation functions:*

$$\mathbb{E}\left[X_{ij}X_{k\ell}\right] = \mathbb{E}\left[Y_{ij}Y_{k\ell}\right] = \frac{1}{n^2}\delta_{ik}\delta_{j\ell}, \qquad\qquad \mathbb{E}\left[X_{ij}Y_{k\ell}\right] = \frac{\rho}{n^2}\delta_{ik}\delta_{j\ell}, \qquad (14)$$

*i.e., $X$ and $Y$ have pairwise i.i.d. entries which are correlated with a Pearson correlation coefficient $\rho$. Assume further that the filter $F \in \mathbb{R}^{r \times r}$ comprises zero-mean i.i.d. Gaussian entries with variance $2/r^2$ and is independent of $(X, Y)$. Then, for a ReLU activation function $\sigma$,*

$$\mathbb{E}\left[\langle \sigma(F * X), \sigma(F * Y)\rangle\right] = \frac{\sqrt{1 - \rho^2} + (\pi - \cos^{-1}(\rho))\rho}{\pi}.$$

### 3.4 SIMPLE BLACK AND WHITE IMAGE MODEL

This section provides insight why we get roughly an isometry when embedding natural images via a random convolution followed by ReLU.

As a conceptual model, we will work with binary pictures $\{0, 1\}^{n \times n}$ or equivalently as subsets of $\mathbb{Z}_n \times \mathbb{Z}_n$. Also, when considering real-life pictures, an apparent characteristic is that they consist of relatively large monochromatic patches, so for the next theorem, one should keep in mind pictures that consist of large convex bodies.

To state our results, we define a notion of a shared $r$-boundary.

**Definition 2** (Boundary). Let $A, B \subset \mathbb{Z}_n \times \mathbb{Z}_n$ and let $r \in [n]$. Then, the $r$-boundary of the intersection between $A$ and $B$, denoted by $\partial_r(A, B)$, is defined as the set of all pixels $(i, j) \in \mathbb{Z}_n \times \mathbb{Z}_n$ such that

$$\exists a_1, b_1, c_1, d_1 \in \{-r, \ldots, r\} : (i + a_1, j + b_1) \in A \land (i + c_1, j + d_1) \in B \qquad (15a)$$
$$\exists a_2, b_2, c_2, d_2 \in \{-r, \ldots, r\} : (i + a_2, j + b_2) \notin A \lor (i + c_2, j + d_2) \notin B \qquad (15b)$$

where the addition in (15a) and (15b) is over $\mathbb{Z}_n$.

In words, a pixel $(i, j)$ belongs to the $r$-boundary $\partial_r(A, B)$ if, considering the square with edge size $2r + 1$ centered at the pixel: (1) the square intersects both $A$ and $B$, (2) the square is not contained in $A$ or is not contained in $B$.

**Example 1.** Let $A$ and $B$ be axis-aligned rectangles of sizes $a \times b$ and $c \times d$, respectively, and an intersection of size $e \times f$ where $e, f \geq 2r$. Then, $|\partial_r(A, B)| = 4r(e + f)$. This is illustrated in figure 2.

The next theorem bounds the inner product between $A$ and $B$ after applying a ReLU convolution layer in terms of the $r$-boundary of the intersection of $A$ and $B$.

**Theorem 5.** *Let $A, B \subset [n] \times [n]$ and a convolution filter $F \in \mathbb{R}^{(2r+1) \times (2r+1)}$ such that the entries in $F$ are i.i.d. Gaussians with variance $2/(2r + 1)^2$. Then,*

$$\langle A, B\rangle - |\partial_r(A, B)| \leq \mathbb{E}\left[\langle \mathrm{R}(F * A), \mathrm{R}(F * B)\rangle\right] \leq \langle A, B\rangle + |\partial_r(A, B)|, \qquad (16a)$$
$$\|A - B\|^2 - 2|\partial_r(A, B)| \leq \mathbb{E}\left[\|\mathrm{R}(F * A) - \mathrm{R}(F * B)\|^2\right] \leq \|A - B\|^2 + 2|\partial_r(A, B)|. \quad (16b)$$

Theorem 5 and corollary 1 imply

$$|\rho_{\text{out}} - \rho_{\text{in}}| \leq \frac{\partial_r(A, B)}{\|A\| \|B\|} + \epsilon$$

with high probability for a large enough number of filters $N$. This shows that a set of images consisting of "large patches", meaning that $\frac{\partial_r(A,B)}{\|A\|\|B\|}$ is small (as in example 1 for small $r$), is embedded almost-isometrically by a random ReLU convolutional layer. Moreover, *any* set of images may be embedded almost-isometrically this way. To see how, fix $r$ while increasing artificially (digitally) the resolution of the images. The latter would yield $\frac{\partial_r(A,B)}{\|A\|\|B\|} \to 0$ as the resolution tends to infinity.

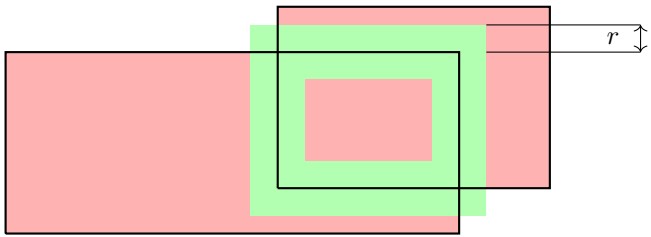

Figure 2: Illustration of the shared $r$-boundary from definition 2 in case of two axis-aligned rectangles. The boundary is marked in green.

## 4 DISCUSSION

### 4.1 GRAYSCALE VERSUS RGB IMAGES

The curious reader might wonder how the analysis over black-and-white (B/W) images can be extended to grayscale and RGB images. While we concentrated on B/W images in this work, the definition of a shared boundary readily extends to grayscale images as does the result of theorem 5. This, in turn, explains the behavior of the grayscale-image dataset F-MNIST in figure 1b.

For RGB images, the framework developed in this work does not hold in general. To see why, consider two extremely simple RGB images: $Red$—consisting of all-zero green and blue channels and a constant unit-norm red channel, and $Green$—consisting of all-zero red and blue channels and a constant unit-norm green channel. Clearly, $\langle Red, Green \rangle = 0$. However, a simple computation, in the spirit of this work, shows that $\mathbb{E}\left[\langle \mathrm{R}(F * Red), \mathrm{R}(F * Green)\rangle\right] = 1/\pi = \widehat{\mathrm{R}}(0)$ which is far from 0. That is, for such monochromatic pairs, the network behaves as a fully connected network (2) and not according to the almost isometric behavior of theorem 5 and figure 1d.

So why does an almost isometric relation appear in figures 1c and 1d? The answer is somewhat surprising and of an independent interest. It appears that for the RGB datasets CIFAR-10 and ImageNet, the RGB channels have high cosine similarity (which might not be surprising on its own), and additionally all the three channels have roughly the same norm. Quantitatively—averaged over $10^4$ images that were picked uniformly at random from the ImageNet dataset—the angle between two channels is ~$12.5°$ and the relative difference between the channel norms is ~$9.2\%$.

This phenomenon implies that our analysis for black and white images (as well as its extension to grayscale images) holds for RGB natural-images datasets. That said, there are images that are roughly monochromatic (namely, predominantly single-color images) but their measure is small and therefore these images can be treated as outliers.

### 4.2 FUTURE WORK

Figure 1 demonstrates that a randomly initialized convolutional layer induces an almost-isometric yet contracting embedding of natural images (mind the linear fit with slope ~$0.97$ in the figure), and therefore the contraction intensifies ("worsens") with each additional layer.

In practice, the datasets are preprocessed by operations such as mean removal and normalization. These operations induce a sharper contraction. Since a sharp collapse in the deeper layers implies failure to achieve dynamical isometry, a natural goal for a practitioner would be to prevent such a collapse. Saxe et al. (2014) propose to replace the i.i.d. Gaussian initialization with orthogonal initialization that is chosen uniformly over the orthogonal group for linear FNNs. However, the latter would fail to prevent the collapse phenomenon as well (see (Pennington et al., 2017)). This can be shown by a similar calculation to that of the dual activation of ReLU with i.i.d. Gaussian initialization. In fact, the dual activations of both initializations will coincide in the limit of large input dimensions.

That said, the collapse can be prevented by moving further away from i.i.d. initialization. For example, by drawing the weights of every filter from the Gaussian distribution under the condition that the filter's sum of weights is zero. Curiously, also including batch normalization between the layers prevents such a collapse. In both cases, the opposite phenomenon to a collapse happens, an expansion. In the deeper layers, the dataset pre-activation vectors become orthogonal to one another. Understanding this expansion and why it holds for seemingly unrelated modifications is a compelling line of research.

ACKNOWLEDGMENTS

The work of I. Nachum and M. Gastpar in this paper was supported in part by the Swiss National Science Foundation under Grant 200364.

The work of A. Khina was supported by the ISRAEL SCIENCE FOUNDATION (grants No. 2077/20) and the WIN Consortium through the Israel Ministry of Economy and Industry.

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

# A  NOTATION

| | |
|---|---|
| $\mathbb{R}, \mathbb{N}, \mathbb{Z}$ | The sets of real, natural, and integer numbers, respectively. |
| $\vee, \wedge$ | The logical 'or' and 'and' operators, respectively. |
| $\mathbb{Z}_n, [n]$ | The sets $\{0, 1, \dots, n-1\}$ and $\{0, 1, \dots, n-1\}$. |
| $A_{ij}, T_{ijk}$ | The $(i, j)$ entry of the matrix $A$ and the $(i, j, k)$ entry of the tensor $T$, respectively. |
| $[T]_{ij}^r$ | The sub-tensor of $T$ formed from rows $i$ to $i + r - 1$ and columns $j$ to $j + r - 1$ (and full third "layer" dimension). |
| $I_k$ | The identity matrix of dimensions $k \times k$. |
| $\mathrm{vec}(\cdot)$ | The vectorization operation of a matrix or a tensor in some systemic order. |
| $\langle x, y \rangle$ | The standard inner product between vectors $x$ and $y$. For matrix or tensor $x$ and $y$ (of the same dimensions) the inner product is defined as $\langle x, y \rangle = \langle \mathrm{vec}(x), \mathrm{vec}(y) \rangle$. |
| $\|x\|$ | The standard Euclidean norm, induced by the standard inner product: $\|x\| = \sqrt{\langle x, x \rangle}$. |
| $\rho$ | The (cosine) similarity. The similarity between $x$ and $y$ is defined as $\rho = \frac{\langle x, y \rangle}{\|x\|\|y\|}$. |
| $\bar{\rho}$ | The mean similarity. $\bar{\rho}$ between random $X$ and $Y$ is defined as $\bar{\rho} = \frac{\mathbb{E}[\langle X, Y \rangle]}{\sqrt{\mathbb{E}[\|X\|^2]\mathbb{E}[\|Y\|^2]}}$. |
| $\sigma, \hat{\sigma}$ | An activation function and its dual, respectively, as defined in definition 1. |
| $\mathrm{R}(x)$ | The ReLU applied to $x$: $\mathrm{R}(x) := \max\{0, x\}$; for a tensor $x$ the R is applied entrywise. |
| $\widehat{\mathrm{R}}(\rho)$ | The dual activation of the ReLU at $\rho \in [-1, 1]$: $\widehat{\mathrm{R}}(\rho) := \frac{\sqrt{1-\rho^2}+(\pi - \cos^{-1}(\rho))\rho}{\pi}$. |
| $\mathcal{N}(\mu, C)$ | A Gaussian vector with mean vector $\mu$ and covariance matrix $C$. |

# B  PROOF OF THEOREM 1

To prove theorem 1, we first prove the following simple result.

**Lemma 3.** *Let $u, v \in \mathbb{R}^k$ be two (deterministic) vectors, and let $G, H \in \mathbb{R}^k$ be zero-mean jointly Gaussian vectors with the following auto- and cross-correlation matrices:*

$$\mathbb{E}\left[GG^T\right] = \mathbb{E}\left[HH^T\right] = \nu^2 I_k, \qquad\qquad \mathbb{E}\left[GH^T\right] = \rho\nu^2 I_k.$$

*Then,*

$$Z = \begin{pmatrix} \langle u, G \rangle \\ \langle v, H \rangle \end{pmatrix} = \begin{pmatrix} u^T G \\ v^T H \end{pmatrix}$$

*is a zero-mean Gaussian vector with a covariance matrix*

$$\mathbb{E}\left[ZZ^T\right] = \nu^2 \begin{pmatrix} \|u\|^2 & \rho\langle u, v \rangle \\ \rho\langle u, v \rangle & \|v\|^2 \end{pmatrix}.$$

*Proof of lemma 3.* $Z$ is a Gaussian vector as a linear combination of two jointly Gaussian vectors. Furthermore,

$$\mathbb{E}[Z] = \mathbb{E}\left[\begin{pmatrix} u^T G \\ v^T H \end{pmatrix}\right] = \begin{pmatrix} u^T \mathbb{E}[G] \\ v^T \mathbb{E}[H] \end{pmatrix} = 0,$$

$$\mathbb{E}\left[ZZ^T\right] = \mathbb{E}\left[\begin{pmatrix} u^T GG^T u & u^T GH^T v \\ v^T HG^T u & v^T HH^T v \end{pmatrix}\right] = \begin{pmatrix} u^T \mathbb{E}\left[GG^T\right] u & u^T \mathbb{E}\left[GH^T\right] v \\ v^T \mathbb{E}\left[HG^T\right] u & v^T \mathbb{E}\left[HH^T\right] v \end{pmatrix}$$

$$= \nu^2 \begin{pmatrix} \|u\|^2 & \rho\langle u, v \rangle \\ \rho\langle u, v \rangle & \|v\|^2 \end{pmatrix} \qquad\qquad \square$$

We are now ready to prove theorem 1.

*Proof of theorem 1.* The equation (7) is proved as follows.

$$\mathbb{E}\left[\langle \sigma(F * x), \sigma(F * y) \rangle\right] = \sum_{i,j \in \mathbb{Z}_n} \mathbb{E}\left[\sigma(\langle F, [x]_{ij}^r \rangle) \cdot \sigma\left(\langle F, [y]_{ij}^r \rangle\right)\right] \tag{17a}$$

$$= \sum_{i,j \in \mathbb{Z}_n} \widehat{\sigma}\left([x]_{ij}^r, [y]_{ij}^r, \nu\right), \tag{17b}$$

where (17a) follows from the definition of the cyclic convolution (4) and the linearity of expectation, and (17b) follows from definition 1 using lemma 3 with $F$ taking the role of $G$ and $H$ (with $\rho = 1$), and $[x]_{ij}^r$ and $[y]_{ij}^r$ taking the roles of $u$ and $v$, respectively.

Now, we set out to prove (8) under the assumption that $\sigma$ is homogeneous and the following notation.

$$\begin{pmatrix} X_{ij} \\ Y_{ij} \end{pmatrix} \sim \mathcal{N}\left( \begin{pmatrix} 0 \\ 0 \end{pmatrix}, \begin{pmatrix} \left\| [x]_{ij}^r \right\|^2 & \langle [x]_{ij}^r, [y]_{ij}^r \rangle \\ \langle [x]_{ij}^r, [y]_{ij}^r \rangle & \left\| [y]_{ij}^r \right\|^2 \end{pmatrix} \right)$$

$$\begin{pmatrix} \bar{X}_{ij} \\ \bar{Y}_{ij} \end{pmatrix} \sim \mathcal{N}\left( \begin{pmatrix} 0 \\ 0 \end{pmatrix}, \begin{pmatrix} 1 & \rho_{ij} \\ \rho_{ij} & 1 \end{pmatrix} \right)$$

$$\mathbb{E}\left[ \langle \sigma(F * x), \sigma(F * y) \rangle \right] = \sum_{i,j \in \mathbb{Z}_n} \widehat{\sigma}\left( [x]_{ij}^r, [y]_{ij}^r, \nu \right)$$

$$= \sum_{i,j \in \mathbb{Z}_n} \mathbb{E}\left[ \sigma(\nu X_{ij}) \cdot \sigma(\nu Y_{ij}) \right] \tag{18a}$$

$$= \nu^2 \sum_{i,j \in \mathbb{Z}_n} \left\| [x]_{ij}^r \right\| \left\| [y]_{ij}^r \right\| \mathbb{E}\left[ \sigma(X_{ij} / \left\| [x]_{ij}^r \right\|) \cdot \sigma(Y_{ij} / \left\| [y]_{ij}^r \right\|) \right] \tag{18b}$$

$$= \nu^2 \sum_{i,j \in \mathbb{Z}_n} \left\| [x]_{ij}^r \right\| \left\| [y]_{ij}^r \right\| \mathbb{E}\left[ \sigma(\bar{X}_{ij}) \cdot \sigma(\bar{Y}_{ij}) \right] \tag{18c}$$

$$= \nu^2 \cdot \nu_\sigma^2 \sum_{i,j \in \mathbb{Z}_n} \left\| [x]_{ij}^r \right\| \left\| [y]_{ij}^r \right\| \widehat{\sigma}\left( \rho_{ij} \right), \tag{18d}$$

where (18a) follows from definition 1, (18b) follows from the homogeneity of $\sigma$, (18c) follows from standard random variable calculus, and (18d) follows from (6).

Finally, to obtain (9), we apply (8) for $x = y$ and use that $\rho_{ij} = 1$ and $\widehat{\sigma}(1) = 1$:

$$\mathbb{E}\left[ \sigma(F * x)^2 \right] = \nu^2 \cdot \nu_\sigma^2 \sum_{i,j \in \mathbb{Z}_n} \| [x]_{ij}^r \|^2 \widehat{\sigma}(\rho_{ij})$$

$$= \nu^2 \cdot \nu_\sigma^2 \sum_{i,j \in \mathbb{Z}_n} \| [x]_{ij}^r \|^2 \widehat{\sigma}(1)$$

$$= \nu^2 \cdot \nu_\sigma^2 \sum_{i,j \in \mathbb{Z}_n} \| [x]_{ij}^r \|^2 = \nu^2 \cdot \nu_\sigma^2 \cdot r^2 \| x \|^2 \ . \qquad \square$$

## C  PROOF OF THEOREM 2

To prove the concentration of measure results of this work, we will make use of the following definition.

**Definition 3.** The Orlicz norm of a random variable (RV) $X$ with respect to (w.r.t.) a convex function $\psi : [0, \infty) \to [0, \infty)$ such that $\psi(0) = 0$ and $\lim_{x \to \infty} \psi(x) = \infty$ is defined by

$$\| X \|_\psi := \inf \left\{ t > 0 \ \middle| \ \mathbb{E}\left[ \psi\left( \frac{|X|}{t} \right) \right] \leq 1 \right\}.$$

In particular, $X$ is said to be *sub-Gaussian* if $\| X \|_{\psi_2} < \infty$, and *sub-exponential* if $\| X \|_{\psi_1} < \infty$, where $\psi_p(x) := \exp\{x^p\} - 1$ for $p \geq 1$.

The following two results, whose proofs are available in (Vershynin, 2018, Ch. 2 and 5.2), will be useful for the proof of theorem 2.

**Lemma 4.** *Let $X$ and $Y$ be sub-Gaussian random variables (not necessarily independent). Then,*

1. ***Sum of independent sub-Gaussians.*** *If $X$ and $Y$ are also independent, then their sum, $X + Y$, is sub-Gaussian. Moreover, $\| X + Y \|_{\psi_2}^2 \leq C \left( \| X \|_{\psi_2}^2 + \| Y \|_{\psi_2}^2 \right)$ for an absolute constant $C$. The same holds (with the same constant $C$) also for sums of multiple independent sub-Gaussian RVs.*

2. **Centering.** $X - \mathbb{E}[X]$ *is sub-Gaussian. Moreover,* $\|X - \mathbb{E}[X]\|_{\psi_2} \leq C \|X\|_{\psi_2}$ *for an absolute constant* $C$.

3. **Lipschitz functions of Gaussians.** *If* $X$ *is a centered Gaussian and* $f$ *is a function with Lipschitz constant* $L$, *then* $\|f(X) - \mathbb{E}[f(X)]\|_{\psi_2} \leq C \|X\|_{\psi_2}$ *for an absolute constant* $C$.

4. **Product of sub-Gaussians.** $XY$ *is sub-exponential. Moreover,* $\|XY\|_{\psi_1} \leq \|X\|_{\psi_2} \|Y\|_{\psi_2}$.

**Theorem 6** (Bernstein's inequality for sub-exponentials). *Let* $X_1, \ldots, X_N$ *be independent zero-mean sub-exponential RVs. Then,*

$$\mathbb{P}\left(\left|\frac{1}{N}\sum_{i=1}^{N} X_i\right| \geq t\right) \leq 2\exp\left\{-\min\left\{\frac{t^2}{K^2}, \frac{t}{K}\right\} \cdot c \cdot N\right\}, \qquad \forall t \geq 0,$$

*where* $K = \max_i \|X_i\|_{\psi_1}$ *and* $c > 0$ *is an absolute constant.*

We are now ready to prove the desired concentration-of-measure result.

*Proof of theorem 2.* $\langle F_\ell, [x]_{ij}^r \rangle$ and $\langle F_\ell, [y]_{ij}^r \rangle$ are (jointly) Gaussian RVs—as linear combinations of i.i.d. Gaussian RVs—with mean zero and variances $\nu^2 \left\|[x]_{ij}^r\right\|^2$ and $\nu^2 \left\|[y]_{ij}^r\right\|^2$, respectively, for all $\ell \in [N]$ and $i, j \in \mathbb{Z}_n$. Hence, $\langle F_\ell, [x]_{ij}^r \rangle$ and $\langle F_\ell, [y]_{ij}^r \rangle$ are also sub-Gaussian with

$$\left\|\langle F_\ell, [x]_{ij}^r \rangle\right\|_{\psi_2} \leq C_0 \nu R, \qquad \left\|\langle F_\ell, [y]_{ij}^r \rangle\right\|_{\psi_2} \leq C_0 \nu R, \qquad \forall \ell \in [N], \qquad \forall i, j \in \mathbb{Z}_n$$

for some universal constant $C_0 > 0$. Define now

$$X_{ij\ell} := \sigma\left(\langle F_\ell, [x]_{ij}^r \rangle\right), \qquad Y_{ij\ell} := \sigma\left(\langle F_\ell, [y]_{ij}^r \rangle\right), \qquad \forall \ell \in [N], \qquad \forall i, j \in \mathbb{Z}_n.$$

Since $\sigma$ is Lipschitz continuous with a Lipschitz constant $L$, and since the inner products $\langle F_\ell, [x]_{ij}^r \rangle$ and $\langle F_\ell, [y]_{ij}^r \rangle$ are centered Gaussians, by property 3 in lemma 4 it follows that $X_{ij\ell}$ and $Y_{ij\ell}$ are sub-Gaussian with

$$\|X_{ij\ell} - \mathbb{E}[X_{ij\ell}]\|_{\psi_2} \leq C_0 L \nu R, \quad \|Y_{ij\ell} - \mathbb{E}[Y_{ij\ell}]\|_{\psi_2} \leq C_0 L \nu R, \quad \forall \ell \in [N], \quad \forall i, j \in \mathbb{Z}_n. \tag{19}$$

Due to the assumption $\sigma(0) = 0$ and the fact that $\sigma$ has Lipschitz constant $L$, inequality $|\sigma(x)| \leq L|x|$ holds for every $x \in \mathbb{R}$. Accordingly, we can bound the expectation of $\mathbb{E}[X_{ij\ell}]$ with

$$|\mathbb{E}[X_{ij\ell}]| \leq \mathbb{E}[|X_{ij\ell}|] \leq L\mathbb{E}\left[\left|\langle F_\ell, [x]_{ij}^r \rangle\right|\right] \leq L\sqrt{\text{Var}\left[\langle F_\ell, [x]_{ij}^r \rangle\right]} \leq L\nu R, \tag{20}$$

where we applied the Cauchy–Schwarz inequality. Similarly, it holds that

$$\left|\mathbb{E}[Y_{ij\ell}]\right| \leq L\nu R. \tag{21}$$

Since by homogeneity of norm it also holds for any constant $D \in \mathbb{R}$ that $\|D\|_{\psi_2} \leq C_0|D|$, by triangle inequality it then follows from (19), (20) and (21) that

$$\|X_{ij\ell}\|_{\psi_2} \leq \|X_{ij\ell} - \mathbb{E}[X_{ij\ell}]\|_{\psi_2} + \|\mathbb{E}[X_{ij\ell}]\|_{\psi_2} \leq 2C_0 L\nu R,$$
$$\|Y_{ij\ell}\|_{\psi_2} \leq 2C_0 L\nu R.$$

Therefore, subsequent application of properties 4 and 2 of lemma 4 to $X_{ij\ell} Y_{ij\ell}$, yields

$$\|X_{ij\ell} Y_{ij\ell}\|_{\psi_1} \leq 4C_0^2 \nu^2 L^2 R^2,$$
$$\|X_{ij\ell} Y_{ij\ell} - \mathbb{E}[X_{ij\ell} Y_{ij\ell}]\|_{\psi_1} \leq 4C_0^2 C\nu^2 L^2 R^2, \tag{22}$$

for all $\ell \in [N]$ and all $i, j \in \mathbb{Z}_n$, where $C > 0$ is is the absolute constant from lemma 4. Now it follows

$$\mathbb{P}\left( \left| \frac{1}{N} \sum_{\ell=1}^{N} \langle \sigma(F_\ell * x), \sigma(F_\ell * y) \rangle - \sum_{i,j=1}^{n} \widehat{\sigma}\left([x]_{ij}^r, [y]_{ij}^r, \nu\right) \right| \geq \epsilon \right)$$

$$= \mathbb{P}\left( \frac{1}{N} \left| \sum_{\ell \in [N]; i,j \in \mathbb{Z}_n} \left\{ X_{ij\ell} Y_{ij\ell} - \mathbb{E}\left[X_{ij\ell} Y_{ij\ell}\right] \right\} \right| \geq \epsilon \right) \tag{23a}$$

$$\leq \mathbb{P}\left( \frac{1}{N} \sum_{i,j \in \mathbb{Z}_n} \left| \sum_{\ell \in [N]} \left\{ X_{ij\ell} Y_{ij\ell} - \mathbb{E}\left[X_{ij\ell} Y_{ij\ell}\right] \right\} \right| \geq \epsilon \right) \tag{23b}$$

$$\leq \mathbb{P}\left( \frac{n^2}{N} \max_{i,j \in \mathbb{Z}_n} \left| \sum_{\ell \in [N]} \left\{ X_{ij\ell} Y_{ij\ell} - \mathbb{E}\left[X_{ij\ell} Y_{ij\ell}\right] \right\} \right| \geq \epsilon \right) \tag{23c}$$

$$\leq \sum_{i,j \in \mathbb{Z}_n} \mathbb{P}\left( \frac{n^2}{N} \left| \sum_{\ell \in [N]} \left\{ X_{ij\ell} Y_{ij\ell} - \mathbb{E}\left[X_{ij\ell} Y_{ij\ell}\right] \right\} \right| \geq \epsilon \right) \tag{23d}$$

$$\leq 2n^2 \exp\left\{ - \min\left( \frac{16\epsilon^2}{C_0^4 C^2 \nu^4 L^4 R^4 n^4}, \frac{4\epsilon}{C_0^2 C \nu^2 L^2 R^2 n^2} \right) cN \right\}, \tag{23e}$$

where (23a) follows from theorem 1 and (19), (23b) follows from the triangle inequality, (23d) follows from the union bound, and (23e) follows from (22) and theorem 6.

Finally, noting that (11) holds for $N > \max\left( \frac{C_0^4 C^2 \nu^4 L^4 R^4 n^4}{16c\epsilon^2}, \frac{C_0^2 C \nu^2 L^2 R^2 n^2}{4c\epsilon} \right) \log \frac{2n^2}{\delta}$ concludes the proof. $\square$

## D   PROOF OF COROLLARY 1

Let us start by inspecting (12) to observe that, by homogeneity of $\sigma$, neither the value of $\bar{\rho}_{\text{out}}$ nor the distribution of $\rho_{\text{out}}$ depend on $\nu$ or the norms of $x$ and $y$. Therefore, let us assume without loss of generality that $x$ and $y$ are unit vectors, as well as that $\nu = \frac{1}{\nu_\sigma r}$. Furthermore, note that homogeneity of $\sigma$ implies $\sigma(0) = 0$, so theorem 2 can be applied with activation function $\sigma$.

Let $A := \frac{1}{N} \sum_{\ell=1}^{N} \langle \sigma(F_\ell * x), \sigma(F_\ell * y) \rangle$, $B := \frac{1}{N} \sum_{\ell=1}^{N} \|\sigma(F_\ell * x)\|^2$, $C := \frac{1}{N} \sum_{\ell=1}^{N} \|\sigma(F_\ell * y)\|^2$ and $A' := \frac{1}{r^2} \sum_{i,j \in \mathbb{Z}_n} \|[x]_{ij}^r\| \cdot \|[y]_{ij}^r\| \widehat{\sigma}(\rho_{ij})$. Comparing against (12), we see that $\rho_{\text{out}} = \frac{A}{\sqrt{BC}}$ and that $\bar{\rho}_{\text{out}} = A'$.

We now apply theorem 2 three times, for pairs of vectors $(x, y)$, $(x, x)$ and $(y, y)$, respectively, with parameters of $\epsilon/5$ and $\delta/3$. Using equations (7)–(9) from theorem 1, we check that indeed each of the three following relations holds for $N > \max(K, K^2) \log \frac{6n^2}{\delta}$, except with probability $\delta/3$:

$$|A - A'| \leq \frac{\epsilon}{5}, \qquad |B - 1| \leq \frac{\epsilon}{5}, \qquad |C - 1| \leq \frac{\epsilon}{5}. \tag{24}$$

By union bound, all three inequalities in (24) hold simultaneously except with probability $\delta$. Finally, by elementary inequalities we establish that if (24) holds for some $A, A', B, C \in \mathbb{R}$ such that $|A| \leq 1$ and $0 < \epsilon \leq 1/10$, then it also holds that

$$\left| \rho_{\text{out}} - \bar{\rho}_{\text{out}} \right| = \left| \frac{A}{\sqrt{BC}} - A' \right| \leq \epsilon.$$

Since this occurs except with probability $\delta$, the proof is concluded. $\square$

# E   PROOF OF LEMMATA 1 AND 2

*Proof of lemma 1.* Lemma 1 may be viewed as a special case of theorem 1 with an identity activation function $\sigma$. Therefore,

$$\mathbb{E}\left[\langle F * x, F * y\rangle\right] \stackrel{(a)}{=} \sum_{i,j\in\mathbb{Z}_n} \widehat{\sigma}\left([x]_{ij}^r, [y]_{ij}^r, \nu\right) \stackrel{(b)}{=} \sum_{i,j\in\mathbb{Z}_n} \nu^2 \langle [x]_{ij}^r, [y]_{ij}^r\rangle \stackrel{(c)}{=} \langle x, y\rangle,$$

where $(a)$ follows from theorem 1, $(b)$ follows from definition 1, and $(c)$ holds since $\nu^2 = 1/r^2$. $\quad\square$

*Proof of lemma 2.* Again, lemma 2 follows from theorem 2 for $\sigma(x) \equiv x$ (and hence $L = 1$) and from lemma 1. $\quad\square$

# F   PROOF OF THEOREM 3

We start with the lower bound. Define $\rho_{ij} := \frac{\langle [x]_{ij}^r, [y]_{ij}^r\rangle}{\|[x]_{ij}^r\| \|[y]_{ij}^r\|}$ for all $i, j \in \mathbb{Z}_n$. Then, for all $i, j \in \mathbb{Z}_n$,

$$\langle [x]_{ij}^r, [y]_{ij}^r\rangle = \left\|[x]_{ij}^r\right\| \left\|[y]_{ij}^r\right\| \rho_{ij} \tag{25a}$$

$$\leq \left\|[x]_{ij}^r\right\| \left\|[y]_{ij}^r\right\| \frac{\sqrt{1 - \rho_{ij}^2} + (\pi - \cos^{-1}(\rho_{ij}))\rho_{ij}}{\pi} \tag{25b}$$

$$= r^2 \mathbb{E}\left[\sigma\left(\langle F, [x]_{ij}^r\rangle\right) \sigma\left(\langle F, [y]_{ij}^r\rangle\right)\right], \tag{25c}$$

where (25a) holds by the definition of $\rho_{ij}$, (25b) holds since

$$a \leq \frac{\sqrt{1 - a^2} + \left(\pi - \cos^{-1}(a)\right) a}{\pi} \qquad\qquad \forall |a| \leq 1,$$

and (25c) follows from (Daniely et al., 2016, Table 1 and Section C of supplement). Thus,

$$\langle x, y\rangle = 1/r^2 \sum_{i,j\in\mathbb{Z}_n} \langle [x]_{ij}^r, [y]_{ij}^r\rangle \leq \sum_{i,j\in\mathbb{Z}_n} \mathbb{E}\left[\sigma\left(\langle F, [x]_{ij}^r\rangle\right) \sigma\left(\langle F, [y]_{ij}^r\rangle\right)\right] = \mathbb{E}\left[\sigma(F * x)\sigma(F * y)\right],$$

$$\tag{26}$$

where the inequality follows from (25), and the last step is due to (17). Now, since the ReLU activation is non-negative, $0 \leq \mathbb{E}\left[\sigma(F * x)\sigma(F * y)\right]$, which completes the proof of the left inequality in (13).

For the upper bound, we use the following convexity argument. First, by homogeneity of ReLU, we can assume without loss of generality that $x$ and $y$ have unit norm. Therefore, it remains to show $\mathbb{E}\left[\sigma(F * x)\sigma(F * y)\right] \leq \frac{1+\rho}{2}$ for $x, y$ such that $\|x\| = \|y\| = 1$ and $\langle x, y\rangle = \rho$.

Recall that $\rho_{ij} = \frac{\langle [x]_{ij}^r, [y]_{ij}^r\rangle}{\|[x]_{ij}^r\| \|[y]_{ij}^r\|}$. We expand the definitions in the same way as in (26) and (25c)

$$\mathbb{E}\left[\sigma(F * x)\sigma(F * y)\right] = \sum_{i,j\in\mathbb{Z}_n} \mathbb{E}\left[\sigma\left(\langle F, [x]_{ij}^r\rangle\right) \sigma\left(\langle F, [y]_{ij}^r\rangle\right)\right]$$

$$= \sum_{i,j\in\mathbb{Z}_n} \frac{\|[x]_{ij}^r\| \|[y]_{ij}^r\|}{r^2} \widehat{R}(\rho_{ij}) . \tag{27}$$

It is easily checked that $\widehat{R}(1) = 1, \widehat{R}(-1) = 0$ and that $\widehat{R}(x)$ is convex for $-1 \leq x \leq 1$. Accordingly, using the decomposition $\rho = \frac{1+\rho}{2} \cdot 1 + \frac{1-\rho}{2} \cdot (-1)$, we have by Jensen's inequality for every $i, j \in \mathbb{Z}_n$

$$\widehat{R}(\rho_{ij}) \leq \frac{1 + \rho_{ij}}{2} .$$

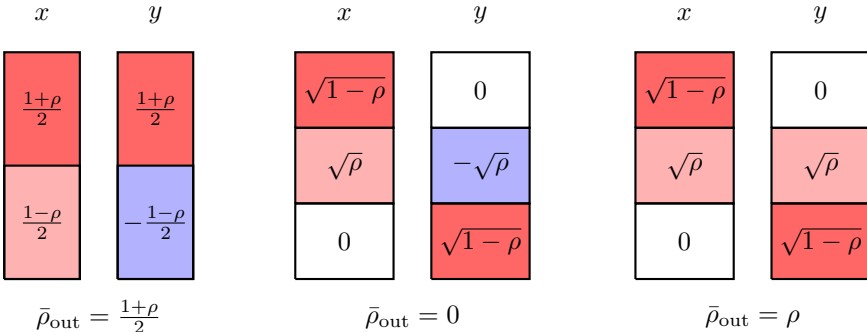

Figure 3: An illustration of the examples from remark 3. Note that similar examples can be constructed in higher dimensions, e.g., by equally distributing the coefficients over a larger set of coordinates.

Substituting into (27),

$$
\begin{aligned}
\mathbb{E}\left[\sigma(F * x)\sigma(F * y)\right] &\leq \sum_{i,j\in\mathbb{Z}_n} \frac{\|[x]_{ij}^r\|\|[y]_{ij}^r\|}{r^2} \cdot \frac{1+\rho_{ij}}{2} \\
&= \frac{1}{2r^2}\sum_{i,j\in\mathbb{Z}_n}\|[x]_{ij}^r\|\|[y]_{ij}^r\| + \frac{1}{2r^2}\sum_{i,j\in\mathbb{Z}_n}\langle[x]_{ij}^r,[y]_{ij}^r\rangle \\
&\leq \frac{1}{2r^2}\sqrt{\left(\sum_{i,j\in\mathbb{Z}_n}\|[x]_{ij}^r\|^2\right)\left(\sum_{i,j\in\mathbb{Z}_n}\|[y]_{ij}^r\|^2\right)} + \frac{1}{2}\cdot\langle x,y\rangle \qquad (28) \\
&= \frac{1+\rho}{2},
\end{aligned}
$$

where in (28) we applied the Cauchy–Schwarz inequality and used the initial assumption $\|x\| = \|y\| = 1$. $\qquad\square$

## G  PROOF OF THEOREM 4

The proof follows from the following steps.

$$
\mathbb{E}\left[\langle\sigma(F * X),\sigma(F * Y)\rangle\right] = \mathbb{E}\left[\mathbb{E}\left[\sum_{i,j\in\mathbb{Z}_n}\sigma(\langle F,[X]_{ij}^r\rangle)\sigma(\langle F,[Y]_{ij}^r\rangle)\,\middle|\,F\right]\right] \qquad (29\text{a})
$$

$$
= \mathbb{E}\left[n^2\mathbb{E}\left[\sigma\left(X_F\right)\sigma\left(Y_F\right)\right]\right] \qquad (29\text{b})
$$

$$
= \frac{1}{2}\frac{\sqrt{1-\rho^2}+\left(\pi-\cos^{-1}(\rho)\right)\rho}{\pi}\mathbb{E}\left[\|F\|^2\right] \qquad (29\text{c})
$$

$$
= \frac{\sqrt{1-\rho^2}+\left(\pi-\cos^{-1}(\rho)\right)\rho}{\pi} \qquad (29\text{d})
$$

where (29a) follows from the law of total expectation; (29b) follows from (14), from the independence of $F$ in $(X,Y)$, and from lemma 3 with $F$ taking the roles of $u$ and $v$, and $[X]_{ij}^r$ and $[Y]_{ij}^r$ taking the roles of $G$ and $H$, resulting in $X_F$ and $Y_F$ which are zero-mean jointly Gaussian with variances $\|F\|^2/n^2$ and a Pearson correlation coefficient $\rho$; (29c) follows from the homogeneity of the ReLU activation function and from (Cho and Saul, 2009, eq. (6)), (Giryes et al., 2016, Theorem 4), (Daniely et al., 2016, Section 8); (29d) holds by the definition of $F$. $\qquad\square$

# H    PROOF OF THEOREM 5

In this section, we will use the following additional notations. $[A]_{ij}^{\bar{r}}$ will denote the submatrix of $A$ formed from rows $i - r$ to $i + r$ and columns $j - r$ to $j + r$, and $\mathbf{1}_{k \times k}$ will denote a $k \times k$ matrix with all entries equal 1.

The main effort in the proof is establishing (16a). We apply theorem 1, so we have $\mathbb{E}\left[\langle \sigma(F * A), \sigma(F * B) \rangle\right] = \sum_{i,j} \widehat{\sigma}([A]_{ij}^{\bar{r}}, [B]_{ij}^{\bar{r}}, 2/(2r+1))$. And specifically, for the ReLU activation, it holds that

$$\widehat{\sigma}([A]_{ij}^{\bar{r}}, [B]_{ij}^{\bar{r}}, 2/(2r+1)) = \frac{\left\|[A]_{ij}^{\bar{r}}\right\| \left\|[B]_{ij}^{\bar{r}}\right\| \widehat{\mathrm{R}}(\rho_{ij})}{(2r+1)^2} \tag{30}$$

where $\rho_{ij} = \frac{\langle [A]_{ij}^{\bar{r}}, [B]_{ij}^{\bar{r}} \rangle}{\left\|[A]_{ij}^{\bar{r}}\right\| \left\|[B]_{ij}^{\bar{r}}\right\|}$.

A natural way to show that the convolutional layer induces isometry in expectation would be to show

$$\widehat{\sigma}([A]_{ij}^{\bar{r}}, [B]_{ij}^{\bar{r}}, 2/(2r+1)) = A_{ij} \cdot B_{ij}$$

for all pairs $(i, j)$. Unfortunately, this does not hold for all pairs. However, we will show that the only pairs for which it does not hold belong to the $r$-boundary $\partial_r(A, B)$. There are two cases to consider $A_{ij} \cdot B_{ij} = 0$ and $A_{ij} \cdot B_{ij} = 1$ since $A$ and $B$ are binary.

Let $\widehat{\sigma}_{ij} = \widehat{\sigma}\left([A]_{ij}^{\bar{r}}, [B]_{ij}^{\bar{r}}), 2/(2r+1)\right)$. Let $E_0 = \{(i, j) \in \mathbb{Z}_n : A_{ij} \cdot B_{ij} = 0 \wedge \widehat{\sigma}_{ij} \neq 0\}$ and $E_1 = \{(i, j) \in \mathbb{Z}_n : A_{ij} \cdot B_{ij} = 1 \wedge \widehat{\sigma}_{ij} \neq 1\}$. Since clearly $0 \leq \widehat{\sigma}_{ij} \leq 1$ and $\langle A, B \rangle = |\{(i, j) : A_{ij} \cdot B_{ij} = 1\}|$, we have

$$\langle A, B \rangle - |E_1| \leq \mathbb{E}_F\left[\sigma(F * A), \sigma(F * B)\right] \leq \langle A, B \rangle + |E_0| .$$

In particular, if we show $E_0, E_1 \subseteq \partial_r(A, B)$, then (16a) will be established. This is what we show in the rest of the proof.

For readability, we repeat here the two conditions for a pair $(i, j)$ to be included in the boundary, see definition 2. Namely, $(i, j)$ belongs to $\partial_r(A, B)$ if and only if both conditions below hold:

1. $\exists a_1, b_1, c_1, d_1 \in \{-r, \dots, r\} : (i + a_1, j + b_1) \in A \wedge (i + c_1, j + d_1) \in B$.
2. $\exists a_2, b_2, c_2, d_2 \in \{-r, \dots, r\} : (i + a_2, j + b_2) \notin A \vee (i + c_2, j + d_2) \notin B$.

Let $(i, j)$ be such that $A_{ij} \cdot B_{ij} = 0$. By equation (30), $\widehat{\sigma}_{ij} = 0$ if and only if $[A]_{ij}^{\bar{r}} = 0$ or $[B]_{ij}^{\bar{r}} = 0$. So, when $A_{ij} \cdot B_{ij} = 0$, for $\widehat{\sigma}_{ij} \neq 0$ to hold it is necessary that there are $a_1, b_1, c_1, d_1 \in \{-r, \dots, r\}$ such that $A_{i+a_1, j+b_1} = 1$ and $A_{i+c_1, j+d_1} = 1$ which corresponds to the first item in definition 2. The second item holds with $a_2 = b_2 = c_2 = d_2 = 0$ since $A_{ij} \cdot B_{ij} = 0$ so either $A_{ij} = 0$ or $B_{ij} = 0$. This shows that pairs of indices for which $A_{ij} \cdot B_{ij} = 0$ and $\widehat{\sigma}_{ij} \neq 0$ are contained in $\partial_r(A, B)$, in other words that $E_0 \subseteq \partial_r(A, B)$.

Now take $(i, j)$ such that $A_{ij} \cdot B_{ij} = 1$. By equation (30), $\widehat{\sigma}_{ij} = 1$ if and only if $[A]_{ij}^{\bar{r}} = \mathbf{1}_{(2r+1) \times (2r+1)}$ and $[B]_{ij}^{\bar{r}} = \mathbf{1}_{(2r+1) \times (2r+1)}$. So, when $A_{ij} \cdot B_{ij} = 1$ and $\widehat{\sigma}_{ij} \neq 1$, then there are $a_2, b_2, c_2, d_2 \in \{-r, \dots, r\}$ such that $A_{i+a_2, j+b_2} = 0$ or $B_{i+c_2, j+d_2} = 0$ which corresponds to the second item in definition 2. The first item holds with $a_1 = b_1 = c_1 = d_1 = 0$ since $A_{ij} \cdot B_{ij} = 1$ so both $A_{ij} = 1$ and $B_{ij} = 1$. This shows that pairs of indices for which $A_{ij} \cdot B_{ij} = 1$ and $\widehat{\sigma}_{ij} \neq 1$ are contained in $\partial_r(A, B)$, or equivalently that $E_1 \subseteq \partial_r(A, B)$.

This concludes the proof of (16a). As for (16b), it follows easily by substituting in the formula $\|A - B\|^2 = \|A\|^2 + \|B\|^2 - 2\langle A, B \rangle$. We then use the fact $\mathbb{E}\left[\left\|\mathrm{R}(F * A)^2\right\|\right] = \|A\|^2$, which follows from (9), recalling that $\nu^2 = 2/(2r+1)^2$ and $\nu_{\mathrm{R}}^2 = 1/2$. $\qquad \square$

