# OpenReview forum: "A Johnson-Lindenstrauss Framework for Randomly Initialized CNNs"
_ICLR.cc/2022/Conference — ICLR 2022 Poster_

### Official Review · Reviewer_DLps · 2021-11-02

**Correctness:** 4
**Technical Novelty And Significance:** 2
**Empirical Novelty And Significance:** Not applicable
**Recommendation:** 6
**Confidence:** 2

**Main Review:**

Strengths:

The paper is well-written. The introduction and related work section give plenty of background information and a notation table is also included to help the reader to access the technique details. I check the technical details and no major flaw is found.


Weaknesses:

The technical strengths may not be strong enough. For the linear CNNs with random Gaussian weights, the underlying procedure is a sum of product between the inputs and random Gaussian random variables. It is expected that the JL lemma-type results will still be held. For the CNNs+ ReLU, since part of the information can be truncted by ReLU, the contraction behavior is also straightforward. If the authors think the proof contains nontrivial contributions, they should be highlighted.

**Summary Of The Paper:**

This paper studies the random initialized CNNs and analyzes the geometry preservation. For linear CNNs, the authors show the JL lemma type results hold. For  CNN+ ReLU, the output contracts and the level of contraction depend on the inputs. In numerical experiments, the authors verify the geometry of natural images is preserved, while for random gaussian correlated inputs, the contraction behavior appears.

**Summary Of The Review:**

My major concern is the technical significance. At the current stage, I think this paper may not be strong enough for ICLR.

**** After seeing the author's comments. I agree the JL-type results for CNNs are different from those in the literature. I change my score accordingly.

---

> ### Author Response · Authors · 2021-11-14
> **Reply to reviewer DLps**
>
> Thank you for the review. Indeed, the fact that *some* form of weak contraction holds is intuitive in line with your explanation. We believe that our main contribution lies in quantifying the contraction. We show that, depending on the character of the dataset, a CNN can show either a strong contraction or almost isometric behavior. This is surprising to us and quite different to the FNNs results of Daniely et al. (2016). [1]
>
> [1] https://arxiv.org/abs/1602.05897

---

### Official Review · Reviewer_g6qL · 2021-11-03

**Correctness:** 4
**Technical Novelty And Significance:** 3
**Empirical Novelty And Significance:** 3
**Recommendation:** 8
**Confidence:** 4

**Main Review:**

The paper is not a typical paper that we read today. The formulation is simple, and the paper is easy to read (in the main part, which does not contain proofs, only formulation).
 Some results are easy to prove (such as Lemma 3 and Theorem 1). Some results have doubtful meaning, such as concentration of measure result (see 22e) for example, which contains very unrealistic constants.
However, the theorems seem to be correct. Also, the authors test their results on different datasets and get intriguing results that show that the lower bound seems to be better suited for random convolutional filter, applied to CIFAR-10 or ImageNet datasets.

A possible addition would be to study how the isometry property "suffers" during the training, but since the authors limit themselves to the analysis of the initialization, that is not formally connected to the results.

Questions:

- After the first CNN filter, the distribution might change. What is the effect of deeper layers on the isometry property?
- Can such kind of studies provide practical recommendations for initialization?


**Summary Of The Paper:**

The paper shows (as clearly described in the introduction) two things: random filters have the distance preservation property. For random convolution + ReLU activation it is contracted (which is also expected, since ReLU throws away half of the entries).
However, for images the authors try to explain why CNN+ReLU is isometric by considering images as objects with large monochromatic images.

The main motivation for such research in practice is initialization. A deep fully-connected network with ReLU may lead to collapse, whereas for CNNs it does not. The paper argues that the randomized theory presented here is a justification for such fact for images.

**Summary Of The Review:**

Good, self-consistent and easy to ready paper. Some proofs are not needed.

---

> ### Author Response · Authors · 2021-11-14
> **Reply to reviewer g6qL**
>
> Thank you for the positive review. We address one of your points and your questions below.
>
> ### Constants in the concentration bounds
>
> As we have replied to the reviewer Mtqu, the constants are not as bad as they may seem at first.
>
> For example, in corollary 1 for inputs $x$, $y$ with entries in $\\{+1, -1\\}$ and ReLU activation, one obtains $K = 4D/\epsilon$. While for simplicity we do not make the constant $D$ explicit, inspecting the proofs of theorem 2 and corollary 1, we do not think it is particularly large.
>
> We further note that the concentration results are conceptually important since they are needed in order to prove that the output similarity on the left in (1), converges to the mean output similarity on the right in (1) by appealing to ergodicity.
>
> ### Deeper layers
>
> This is of course a very good question, and the answers can be complicated.
> Perhaps good examples are the F-MNIST/CIFAR/ImageNet pictures and the model of bitmaps with small boundaries in section 3.4 that is supposed to reflect them.
>
> Empirically, each additional layer seems to introduce a small additional contraction in the embeddings of the dataset. This is consistent with the fact that theorem 5 gives only an approximate isometry, and that our “large patch” model only approximately reflects the structure of real datasets. For example, for a pair of images from F-MNIST separated by an angle $\alpha$ typically the angle between the embeddings will be around $0.97 \alpha$ for one layer, and $(0.97)^{10} \alpha \approx 0.74 \alpha$ for 10 layers. In the extreme cases of very deep (too deep?) networks, the embeddings become concentrated around a single point.
>
> On the theoretical side, we believe a more general version of theorem 5 can be proved. That version would be valid for “large patch” pictures in the regime with small filter size $r$ and growing picture size $n$, with constant factors that depend on the depth $d$.
>
> We can expand on these points if requested. For reasons of space and simplicity we decided against discussing them in the submission.
>
> ### Practical implications for initialization
>
> This is also a very valid question.
> As we discuss in section 4.2, there are some indications that batch normalization prevents an excessive contraction in the embeddings. We also suspect that, for small filters, it might be helpful to sample the weights from the i.i.d. distribution conditioned on the sum of the weights equal to zero. However, these are preliminary thoughts and we are not quite ready to include a more extensive discussion.

---

### Official Review · Reviewer_Mtqu · 2021-11-05

**Correctness:** 3
**Technical Novelty And Significance:** 3
**Empirical Novelty And Significance:** 3
**Recommendation:** 8
**Confidence:** 4

**Main Review:**


### Strengths

- The paper is well-written and enjoyable to read. There is a good mix of
  theory and intuition. The mathematics is written clearly and precisely.
- The basic observation around angle contraction in random convolution layers,
  while possibly implicit in some previous works (see request for clarification
  below), does not seem to have been presented systematically as the authors do
  so here, and is quite an interesting difference relative to the feedforward
  case. The way the authors study this issue, using tight lower and upper
  bounds for the new map with the ReLU activation, as well as examples with
  gaussian data, empirical data (Figure 1) and cartoon data (section 3.4), is
  sufficiently systematic to give a good entry point to understand various
  interesting aspects of this difference. I anticipate this will be useful for
  follow-up work.
- The issue is placed well in context via discussion of many related works
  about dynamical isometry and more generally accelerating training of
  deep networks.


### Weaknesses
- There is a large body of work on CNTKs that has appeared following the
  initial interest in the NTK, and I think it would be helpful if the authors
  spent some time in the related work section comparing to any relevant work in
  this line -- it seems to me, although I could be mistaken, that some of the
  formulas and concentration results the authors give in the paper may have
  been derived previously in these contexts due to the natural interest in such
  expressions for writing down CNTKs, and it is important to know of such
  overlaps to assess the authors' work's novelty. For example, [1-4] below.
- The concentration results are rather primitive: notably Theorem 2
  requires a stack of i.i.d.\ feature maps to be channel-averaged to obtain
  concentration, whereas one would expect it is possible to have a certain
  degree of concentration as long as the input size $n$ is sufficiently large
  relative to the kernel size $r$. This weakness is due to the authors using
  worst-case analysis of the convolutional structure in Theorem 2's proof -- a
  union bound is taken to control the concentration of the individual
  translates of the filter rather than something more sophisticated like
  chaining/decoupling tools, which are standard in the sharp analysis of
  RIP-type properties of convolutional operators (e.g. [5-6]). In this
  connection, the content of Section 3.1 seems superfluous. There also exist in
  the signal propagation literature significantly sharper analyses of the
  concentration behavior of the ReLU's dual activation that are relevant in the
  context of results like Corollary 1 applied to the ReLU [7, Lemma E.3]. All
  this said, I do not believe this necessarily diminishes the authors' results,
  as long as the previous point about comparisons to the literature is
  satisfied -- if this work is deriving these kinds of results for the first
  time in the literature, they of course do not need to be optimal -- but it is
  necessary to acknowledge these limitations to accurately place the results in
  context and make follow-up directions clear.
- Proof of theorem 2: the argument around the Lipschitz property is not
  correct: it is not true in general that a Lipschitz function of a
  sub-gaussian random variable is sub-gaussian (consider a Rademacher random
  vector). The argument should use the gaussian distribution of the random
  variables (and Gauss-Lipschitz concentration e.g. from Vershynin ch. 5 should
  be cited).

[1] http://arxiv.org/abs/1905.12173

[2] http://arxiv.org/abs/1904.11955

[3] http://arxiv.org/abs/2102.10032

[4] http://arxiv.org/abs/1806.05393

[5] http://dx.doi.org/10.1002/cpa.21504

[6] http://arxiv.org/abs/1712.00716

[7] https://openreview.net/forum?id=O-6Pm_d_Q-


### Minor points:
- I don't immediately understand the statement "the unit vectors $x, y$ are
  $1$-dimensional" in Remark 2 -- is this thinking of $x$ and $y$ as being $1
  \times 1 \times 2$ images? In the next sentence we have $x, y \in \mathbb{R}^2$
  stated, i.e. not $1$-dimensional.
- It would be helpful to have some visualizations of the different regimes of
  behavior represented in the bounds of Theorem 3, as well as some more insight
  into the examples of Remarks 2 and 3. For example, can we understand when
  this bound is loose/tight based on the structure of the inputs across
  patches? (The gaussian example recovers the FFNN case -- when there are
  correlations across patches, does this lead us towards the other extremes?)
- First paragraph on page 3: typo "outout" (output)
- The last two paragraphs of the introduction (Section 1) are slightly hard to
  appreciate to this reader because they are written in what seems to be a
  deliberately technically imprecise way -- for example, it would be helpful to
  specify what kinds of depth-width and initialization regimes are being
  considered here (the discussion quickly jumps from angle propagation across
  one layer to angle propagation throughout many layers -- to have the points
  being made here have meaning, I believe it is necessary to be considering
  "edge of chaos" initialization). I particularly do not understand the last
  paragraph unless it is being assumed here that the depth is significantly
  larger than the width (e.g. width fixed as depth diverges).  I think the
  point made in the second to last paragraph is insightful and applies much
  more generally than the "diverging depth" setting.
- Corollary 1: the statement says the variance of the gaussian is $\nu^2$, but
  it seems from the value given for $K$ that a specific choice of $\nu$ is
  being used here.
- Theorem 5: Above the $r$-boundary is defined as a set -- is the usage here
  supposed to be the cardinality of the $r$-boundary?
- section 4.1: "curios"


**Summary Of The Paper:**


The authors consider angle propagation in randomly-initialized convolutional
neural network layers -- given two input images that have cosine angle $\rho$,
what is the cosine angle of their feature embeddings when propagated through
the random layer? They show that the behavior is very different from the
standard feedforward case that has been discussed at length following the work
of Daniely et al. (2016): for filters of small spatial support and e.g. ReLU
activations, the output cosine angle can be as small as the ReLU of the input
cosine angle, or as large as a linear function of the input cosine angle,
depending on the patch structure of the input image. This has implications for
understanding signal propagation and issues like dynamical isometry in
convolutional networks. The authors give some measure concentration results, in
the spirit of Daniely et al.'s results for feedforward networks, that specify
the behavior in the convolutional case, and present examples of different types
of images (unstructured/gaussian; cartoons; CIFAR-10 data) where different
angle propagation behaviors are observed.


**Summary Of The Review:**

This is an interesting paper that combines theory and empirical insights to
highlight differences between angle propagation in random CNNs and feedforward
networks. The results should inspire interesting follow-up work in this
setting. I hope the authors will clarify the relationship to the theorems in
this work to any possible prior art in e.g. CNTK papers, as mentioned above.
The concentration results are basic, but probably enough as first-cuts (it
seems important to acknowledge limitations, however).

---

> ### Author Response · Authors · 2021-11-14
> **Reply to reviewer Mtqu**
>
> Thank you for the thorough and helpful review, and for the literature pointers. We appreciate the nice intuition of thinking of our model from section 3.4 as “cartoons”.
>
> ### Novelty and related literature
>
> Thank you for pointing out the references. We agree that they are relevant and we have added an appropriate discussion to the paper. In short, while there is definitely some thematic overlap, we believe our main insights are novel. In more detail:
> * Theorem 1 may be deduced as a special case of the previously obtained results for CNTKs in [1]–[3]. We added a statement about this after Theorem 1 in the new version.
> * That said, we are not aware of previous concentration results that apply to theorem 2. There are some related bounds, but in different settings (usually fully-connected rather than convolutional layers). As the reviewer rightly states, our proof utilizes known techniques.
> * More importantly, we believe the geometric questions and insights from sections 3.2–3.4 are novel.  Specifically, theorems 3-5 are new: theorem 3 describes to what extent the embedded inner products can increase, while theorems 4 and 5 reveal how inner products of specific datasets propagate through a convolutional layer.
>
> ### Quality of the concentration bounds
> While our concentration bounds are indeed basic, we argue that they are not as bad as it might seem. For example, consider corollary 1, which is an application of theorem 2 to the normalized notion of similarity (we prefer to look at that corollary to avoid confusion related to scaling; a version of this remark applies also directly to theorem 2). Let $x$ and $y$ be inputs with entries in $\\{+1, -1\\}$. Then, calculating the constant in corollary 1, we have $R = r/n$ and $K = D L^2 / \epsilon \nu_\sigma^2$, which does not depend on $n$. The only remaining effect of the union bound is in the more modest $\log(n^2)$ factor.
>
> We have added a remark (remark 2 in the new version) after corollary 1 to clarify this issue.
>
> We don’t think that the intuition that $r \\ll n$ is sufficient for concentration is entirely correct. For example, consider two “cartoon” pictures with small boundaries as in theorem 5. The small boundary condition ensures that most of the $r \\times r$ windows in the inputs are either all-ones or all-zeros. Therefore, due to weight sharing, the deviations for a single filter are highly correlated, and multiple filters need to be applied to achieve concentration, regardless of the values of $r$ and $n$. Admittedly, the bound is still loose due to the $\log(n^2)$ factor in the union bound and the fact that a better concentration for ReLU could be invoked.
>
> We also added a reference to [5] after our lemmas 1 and 2.
>
> Thank you for pointing out the references, and in particular the sharp angle concentration for ReLU in (Buchanan et al. 2021, Lemma E.3) [4]. We have added a relevant remark in section 3.
>
> ### Other issues
>
> * Section 3.1 is included mostly to present a complete picture.
> * **Lipschitz issue in the proof of theorem 2.** Thank you for spotting this. Indeed there was a problem at this step that we now fixed. Please note that we now added an additional assumption $\sigma(0)=0$ on the activation function.
> * In remark 2 (now remark 3), we meant that, for simplicity, $x$ and $y$ are vectors rather than matrices or tensors. We have clarified that in the remark. We have also fixed the other typos, thank you for finding them.
> * **Graphical illustration of remark 2.** We have added a graphical illustration of the examples from the former remark 2 (see figure 3). We hope that this helps with understanding the structure of larger, similar examples.
> * **Dynamical isometry.**
>   * Our remark about collapsing outputs applies in expectation for any relation of depth to width. In general, increasing the width will make the concentration better and the collapse more likely. However, we do not attempt to quantify it in this literature discussion. For a related discussion, see the discussion around Claim 1 in (Daniely et al., 2016) [4].
>   * In this paragraph, we discuss a ReLU FNN at initialization and how angles evolve. Due to homogeneity of ReLU, collapse will occur independent of any initialization variance. So we think that the “edge of chaos” regime is not relevant to our discussion.
>   * We have also updated this paragraph in the submission for increased clarity.
> * **Statement of corollary 1.** We believe that the statement of corollary 1 is correct. Corollary 1 bounds the deviation in cosine similarity, which is a normalized quantity as opposed to the inner product in theorem 2. Therefore, the constant $K$ does not depend on the initial variance $\nu$ (note that the corollary has a stronger assumption of homogeneous activation).
>
> [1] https://arxiv.org/abs/1904.11955
>
> [2] https://arxiv.org/abs/1905.12173
>
> [3] https://arxiv.org/abs/2102.10032
>
> [4] https://arxiv.org/abs/2008.11245
>
> [5] https://arxiv.org/abs/1207.0235

---

### Author Response · Authors · 2021-11-14
**General response**

We thank the reviewers for their consideration. We address specific concerns in our replies to the individual reviewers. We have also uploaded an updated version of our submission with the changes highlighted in red. We discuss the changes in our replies.

As discussed in the individual reply, reviewer Mtqu has spotted an issue in the proof of theorem 2. We have fixed this problem by adding an assumption $\sigma(0)=0$ to theorem 2.

---

> ### Author Response · Authors · 2021-11-19
> **Further response**
>
> We modified the paper to fit the main text into 9 pages. We tried addressing all the points raised by the reviewers. If we failed to fully address some of the comments or concerns raised by the reviewers, we will appreciate receiving feedback from the reviewers to improve the manuscript before the end of the discussion period.

---

### Decision · Program_Chairs · 2022-01-20

**Decision:**

Accept (Poster)

**Comment:**

This paper focuses on understanding how the angle between two inputs change as they are propagated in a randomly-initialized convolutional neural network layers. They demonstrate very different behavior in different settings and provide rigorous measure concentration results. The reviewers thought the paper is well written and easy to read with nice theoretical results. They did raise a variety of technical concerns that were mostly addressed by the authors rebuttal. My own reading of the paper is that this is a nice contribution. I therefore agree with the reviewers and recommend acceptance.